# Digital Media for Behavior Change: Review of an Emerging Field of Study

**DOI:** 10.3390/ijerph19159129

**Published:** 2022-07-26

**Authors:** William Douglas Evans, Lorien C. Abroms, David Broniatowski, Melissa Napolitano, Jeanie Arnold, Megumi Ichimiya, Sohail Agha

**Affiliations:** 1Milken Institute School of Public Health, The George Washington University, 950 New Hampshire Avenue NW, Washington, DC 20052, USA; lorien@gwu.edu (L.C.A.); broniatowski@gwu.edu (D.B.); mnapolitano@gwu.edu (M.N.); jeaniearnold@gwmail.gwu.edu (J.A.); imegumi7@gwmail.gwu.edu (M.I.); 2The BRIGHT Institute, The George Washington University, 950 New Hampshire Avenue NW, Washington, DC 20052, USA; 3Stanford Behavior Design Lab, Seattle, WA 98109, USA; sohailagha@gmail.com

**Keywords:** digital media, behavior change, public health, health care, intervention research, evaluation, social media, self-management, dose-response analysis

## Abstract

Digital media are omnipresent in modern life, but the science on the impact of digital media on behavior is still in its infancy. There is an emerging evidence base of how to use digital media for behavior change. Strategies to change behavior implemented using digital technology have included a variety of platforms and program strategies, all of which are potentially more effective with increased frequency, intensity, interactivity, and feedback. It is critical to accelerate the pace of research on digital platforms, including social media, to understand and address its effects on human behavior. The purpose of the current paper is to provide an overview and describe methods in this emerging field, present use cases, describe a future agenda, and raise central questions to be addressed in future digital health research for behavior change. Digital media for behavior change employs three main methods: (1) digital media interventions, (2) formative research using digital media, and (3) digital media used to conduct evaluations. We examine use cases across several content areas including healthy weight management, tobacco control, and vaccination uptake, to describe and illustrate the methods and potential impact of this emerging field of study. In the discussion, we note that digital media interventions need to explore the full range of functionality of digital devices and their near-constant role in personal self-management and day-to-day living to maximize opportunities for behavior change. Future experimental research should rigorously examine the effects of variable levels of engagement with, and frequency and intensity of exposure to, multiple forms of digital media for behavior change.

## 1. Introduction

### 1.1. Brief Summary of Current Digital Health Research

Digital media (i.e., electronic media where data are stored in digital form) are omnipresent in modern life, but the science of how digital media impact behavior is still in its infancy [1]. For example, approximately 45% of the world’s population or 3.5 billion people use social media, with the average user spending approximately 3 h of their day overall on social media [2]. These statistics make it critical to understand both how technologies such as social media influence health decision making and behavior and to design and evaluate effective behavior change interventions using social and other digital media platforms (i.e., programs aimed at changing specific behaviors in a population using one or more digital platforms as the delivery channel, including research aimed at assessing the effectiveness of such programs).

There is an emerging evidence base of how to use digital media for behavior change. Strategies to change behavior implemented using digital technology have included a variety of platforms (e.g., text messaging, social media, apps) and program strategies (e.g., social media support groups and tailored coaching), all of which are potentially more effective with increased frequency, intensity, interactivity, and feedback [3]. Overall, the most effective health behavior change interventions use a combination of both digital and face-to-face components, lending credence to the importance of classical social behavior change modalities, including human interaction and in-person accountability [1,4].

The most commonly cited research gap is the use of inconsistent, non-standardized measures (e.g., engagement, reach) to evaluate digital media-related behavior change interventions [5,6]. Other areas highlighted for improvement include clarification of exposure and dose, intensity of intervention delivery, and measurement of long-term outcomes [7]. As noted in multiple systematic reviews, the preponderance of evidence characterizing effective behavior change techniques using digital interventions has been collected through focusing on residents of high-income countries (HICs) [8].

What is the distinctive approach in this domain? The field of digital media for behavior change is characterized by its use of digital device-human interactions as an intervention strategy, as a methodology for data collection and research, and as environmental influences (i.e., the study of infodemiology) that may affect behavior and moderate the effects of interventions aimed at behavior change [9]. This paper focuses primarily on the first of these two approaches and explores case studies to illustrate how they have been applied and studied in recent research aimed at identifying effective strategies to bring about positive, population-level health behavior change.

In fact, social media is a core feature of the current social environment, and researchers need to study it to identify media effects, both positive (potential as a behavior change intervention platform) and negative (the effects of mis- and dis-information on decision-making and health behaviors). For example, social media facilitates access to communities around maladaptive health behaviors (e.g., restrictive or binge eating related to anorexia/bulimia), access to networks for illegal purchases such as guns and illegal drugs; its design features (e.g., such as design of News Feed) affect not only health information, but norms and sense of social support [10]. Social media use has been associated with mental health conditions including loneliness and depression, especially among adolescents and young adults [11]. It is critical to accelerate the pace of research on digital platforms, including social media, to understand and address its effects on human behavior.

### 1.2. Overview of Digital Interventions

As an intervention strategy, digital media for behavior change uses all features of digital devices to communicate and create environmental cues and incentives (i.e., following behavioral economics) that encourage the adoption of new behaviors and the maintenance of existing ones. These approaches may take place on social media, mobile phone apps, chatbots, text, and social messaging (e.g., WhatsApp), as well as many specific modalities of information that can be communicated within them, such as video, memes, website links, and graphic images, among others. Digital interventions have the flexibility to be based on tailored, individual communication and on group-level communication, such as in a Facebook group, enabling group interaction and social support. Peer-to-peer interventions (where the intervention is essentially delivered by the participants) are also possible [12].

Digital platforms such as social media have the inherent feature of interactivity and the potential to engage participants and populations in the context of their social networks, thus building a sense of identification and connection with the intervention. This can be done through ‘gamification’ features, such as virtual incentives and rewards and through social role modeling by individuals that are appealing and aspirational for the audience. Digital platforms, like social media, provide the opportunity for participant co-creation (i.e., content co-generated by users and investigators). For example, in the US, adolescents developed and disseminated their own sexual health, substance use, and violence risk behavior prevention messages as part of a community-based participatory Latino and Immigrant health intervention [13].

### 1.3. Examples of Digital Health Research

As a data collection and research strategy, digital media may be used in many ways. Social media provides large quantities of available data on registered users (e.g., Facebook analytics) that can be used to identify potential study participants (e.g., individuals who, based on Facebook data, are likely to fit a specific socio-demographic profile, such as young male Latinos). Apps such as Facebook Messenger and other ‘chat’ functions, may be automated to run surveys through ‘chatbots’ that deliver questions through individual messages with pre-defined response options for study participants. Such technologies can be used to design randomized controlled trials [14].

Additionally, intervention research may be conducted using the combination of social media technology for remarketing (i.e., delivery of specific content, such as advertising, to an individual user based on previous online activity, such as viewing social media content) and chatbot data collection. Therefore, participants may be recruited into a research study using Facebook advertising (e.g., to join a research study), the data are then collected using chatbot technology, and then the same participants may be exposed to an intervention using remarketing technology, and subsequent data may be collected. This creates the potential for social media based randomized experimental studies to evaluate online behavior change interventions.

Behavior change interventions that rely on social networks for their success are hypothesized to have greater impact, and to generate greater interactivity and feedback, than interventions that rely on changes in individual behavior, due to the amplifying effects of social support and social participation [15,16]. Indeed, social network researchers have measured the impact of interventions beyond those immediately exposed to an intervention. Thus, in addition to the effect of treatment-on-the-treated, social network researchers have measured the effect of treatment on the untreated (e.g., for healthy weight management and weight loss) [17]. Researchers who have evaluated door-to-door campaigns to get out the vote, for example, have measured a smaller, discernable effect of such campaigns on the untreated (i.e., effects on members of the household who are not directly exposed to the canvasser) [18]. This additive effect may be due to effects on social norms and other social influences (e.g., based on the observation of those who receive treatment by those who do not).

### 1.4. Evidence on Digital Media for Behavior Change

A recent study illustrates the state of the evidence for digital media behavior change interventions. In this scoping review, the authors found over 3300 articles that met inclusion criteria such as (1) using some form of digital media in an intervention, and (2) having a focus on behavior change or change in antecedents of behavior [19]. This review aimed to (1) routinely monitor and identify literature around digital media-based behavior change interventions to get insights for future digital media studies, and (2) systematically review relevant literature to identify evidence and areas for improvement in the field. Initial findings confirmed that there is emerging published data on exposure to and evaluations of digital health behavior change interventions, but this research is still in its infancy. However, only some 298 reported original research aimed at establishing the effectiveness of a digital behavior change intervention.

The purpose of the current paper is to describe methods in the field, present illustrative use cases, describe a future agenda, and raise central questions to be addressed in future digital health research for behavior change. The science of digital health and its application to promoting health and preventing disease is in its infancy. But new technologies and opportunities to apply digital media technology for behavior change (e.g., how social media, perhaps the most widespread digital media channel, can be used to promote healthy behaviors) are growing rapidly. Social and behavioral scientists must harness the potential of these technologies and develop this emerging field of study. This paper lays out key considerations in that endeavor informed by the extant literature.

## 2. Methods

Digital health for behavior change employs both intervention and research methodologies. In the context of behavior change interventions, there are three main methods: (1) digital media interventions aimed at behavior change, (2) formative research using digital media aimed at aiding in the design of interventions, and (3) digital media used to conduct outcome and impact evaluations. We note that these are methods for conducting digital media practice and research. Below, we provide detailed use cases on the implementation (or results) of such methods.

### 2.1. Examples of Methods to Deliver Digital Media Campaigns for Behavior Change

On social media (the predominant digital communication channel), interventions can be delivered in a variety of ways. Social media (e.g., Facebook, Twitter) can be used as a digital environment for serving ads that can be microtargeted to the individual characteristics of users. Additionally, social media accounts with many followers (e.g., influencers) have been used to promote behavior change messaging with their followers. Social media platforms can also serve as a platform for group-based interventions aimed at behavior change. Several large RCTs have tested the efficacy of randomizing people into private social media groups and then using the groups as the settings for providing education content to users in the form of group posts, including stimulating engagement among users (e.g., polls, questions, conversations) and social support among group members [20,21,22]. These can be supplemented with direct messages to individual users providing additional individualized support or information. Increasingly, there is also an interest in shaping the content of existing groups and pages on social media by having lay health workers either join such groups and post health-promoting content or reach out to group and page administrators requesting them to post such content.

These strategies have potential to create a sense of widespread support for and adoption of specific behaviors. A kind of bandwagon effect may result, leading to behavior change. Theoretically, the effect of such social media campaigns may be to promote a social norm in support of specific health behaviors, such as COVID-19 vaccination, healthy eating and physical activity, or avoidance of nicotine consumption [19]. The following case studies illustrate interventions and research that investigate this hypothesized effect of social media campaigns.

### 2.2. Examples of Digital Media Methods in Formative Research

Another important research opportunity and capability while using digital media is to conduct formative research, or research aimed at helping to design campaigns for behavior change. One example of such efforts was a formative study conducted by Agha and colleagues in Nigeria in 2021. This study applied a behavioral lens to understand drivers of COVID-19 vaccination uptake among healthcare workers (HCWs) in Nigeria. The study used data from an online survey of Nigerian HCWs ages 18 and older conducted in July 2021. Analyses examined the predictors of getting two doses of a COVID-19 vaccine. One-third of HCWs in this study reported that they had gotten two doses of a COVID-19 vaccine. Motivation and ability were powerful predictors of being fully vaccinated: HCWs with high motivation and high ability had a 15-times higher odds ratio of being fully vaccinated. However, only 27% of HCWs had high motivation and high ability. This was primarily because the ability to get vaccinated was quite low among HCWs: only 32% of HCWs reported that it was very easy to get a COVID-19 vaccination. By comparison, motivation was relatively high: 69% of HCWs reported that a COVID-19 vaccine was very important for their health. Much of the recent literature coming out of Nigeria and other LMICs focuses on increasing motivation to get a COVID-19 vaccination. Findings highlight the urgency of making it easier for HCWs to get COVID-19 vaccinations.

The Agha et al. study is an example of how research using digital media, and specifically social media platforms, has tremendous potential to provide insight into behavior change campaign design and implementation. Findings from this study informed a large scale COVID-19 vaccination campaign that began in Nigeria in late 2021 to continue through early 2023 [CITE]. The purpose of this effort is to use social media delivered by trusted organizations and influencers to promote HCW vaccination and broader population vaccination, in part through the role modeling effects of increased rates of vaccinated HCW. This approach has broad applicability to other social media-based research.

### 2.3. Examples of Digital Media Methods for Evaluation Research

Finally, the outcomes and impact of digital media interventions for behavior change may be evaluated using digital media and social media platforms. The previously mentioned example of virtual lab is a state-of-the-art example in the social media domain. The power of social media to first identity individual users who appear, based on publicly available data, to fit a specific demographic, behavioral, or lifestyle profile (e.g., being a health care provider in a specific country of a certain age range) allows for targeting of recruitment efforts. Facebook advertising, for example, may be used to reach a specified population based on Facebook proprietary data, and then those individuals may be invited through the ads to join a research study. Upon initial expression of interest through the clicking on a link to a survey, additional screening may be done (e.g., through eligibility questions) to confirm study inclusion or exclusion, followed by informed consent.

Such studies may follow multiple designs, including observational, quasi-experimental (e.g., examining the effects of exposure to social media messages on outcomes of interest, such as vaccine hesitancy or vaccination) and randomized controlled studies. In social media, the Facebook Messenger (DM) app is one relatively simple way to deliver surveys, as individual questionnaire items may be delivered as chats DMs in sequence to allow a participant to complete a survey wave. Such surveys may be followed by randomization to study condition to receive a social media and/or other treatment aimed at promoting behavior change and/or other outcomes (e.g., changes in social norms or intentions). Because the participants have provided contact data through the social media platform (e.g., through DM), they may be recontacted for longitudinal data collection. In this way, bespoke panels may be created to run studies. Additionally, surveys, as individual questionnaire items, can be delivered with text messaging and apps, offering real-time assessments (i.e., ecological momentary assessments, EMA) that can monitor time sensitive symptoms or conditions such as urges and cravings related to addiction.

An important advantage of this methodology and technology is that it is relatively low cost [23]. Large scale data collection may be conducted with relatively low marginal costs of initial survey programming and data management. Incentives may be provided through such modes as electronic gift cards or mobile phone use credits, the latter being highly valuable in many low- and middle-income countries (LMIC) where pay-by-use mobile phone plans are common. The following use cases illustrate examples of how these methodologies have been recently applied, and we explore the potential of social media platforms in the future development of digital health research.

## 3. Results

### 3.1. Use Cases

In this section, we briefly describe several use cases and their contribution to building the field of digital health for behavior change. These case studies represent results in that they illustrate the application and range of methods in the field, recognizing that there are many other forms that such interventions and research may take.

#### 3.1.1. Use Case #1: Social Media to Deliver Weight Loss Information to Young Adults on University Campuses

Nearly 40% of young adults, ages 20–39 [24], have overweight and obesity. Young adulthood is a pivotal life period, marked by changes in academic and social functioning, food environment and availability, as well as declines in physical activity [25,26]. The availability of weight loss services on university campuses has been delayed in implementation in comparison to other health-risk behaviors [27,28], thus making this an important target for scalable, easy-to-implement interventions. The Healthy Body Healthy U (HBHU) project was designed to examine the differential efficacy of two digitally delivered weight loss treatments compared with a health education control [29]. We will discuss the use of social media for recruitment, consent processes specifically focused on using a commercially available platform (i.e., Facebook) or research purposes, ongoing monitoring and safety procedures, and ways to measure impact.

Participants were young adults (18–35 years) with a body mass index (BMI) of 25–45 kg/m^2^ who attended a university within the Washington, DC or Boston area. Additional inclusion criteria specified that those eligible must be an active Facebook user (i.e., having logged into the platform at least in 1 time within the last month), able to send/receive text messages and no health contraindications for participating in physical activity or weight loss.

There were different channels for recruitment using social marketing principles [30], such as placement of materials in high-trafficked areas and standardized branding using study-specific colors, logos, and fonts. Furthermore, technology was used as a specific outreach strategy. The following figures illustrate how these elements of the program were implemented. “Micro targeting” or delivering advertisements through Facebook targeting specific demographics, was used (see Figure 1). Key leaders on campus, such as academic deans, also tweeted or retweeted study-related recruitment efforts (see Figure 2).

After participants responded to the study advertisements, they completed screening to determine eligibility to participate in the study, at which point they were scheduled for an in-person assessment to further review the study commitment and consent to participate [29]. The consent forms highlighted the potential risks of participating in an online group through a commercial platform with unknown follow-participants. We followed procedures as suggested by Moreno and included the following in the consent document [31]: “*… while the study team will keep your study information confidential, ** anything you post on Facebook is technically governed by and can be used by Facebook**; therefore, we cannot ensure complete confidentiality of all of your Facebook posts and information*” and referred the participant to periodically review the Facebook terms of conditions. We also highlighted the community guidelines and limits to confidentiality based on the presence of others in the group, i.e., “*Confidentiality of your identity or information discussed on the Facebook group page cannot be guaranteed by the research staff due to the presence of other research participants. Study volunteers are **encouraged to maintain strict confidentiality regarding your information and the information of others who are participating in the program***”.

Briefly, the interventions were delivered via private Facebook and SMS text messaging. The participants (*n* = 459) were randomly assigned to one of two weight loss groups delivered via these channels or a health education contact control. The two weight loss interventions were based on the Diabetes Prevention Program [32,33]. The two weight loss interventions differed in the amount of personalization provided. Specifically, one group (tailored) received personalized information based on their own feedback and high-risk barriers, while the other (targeted) received generic weight loss information specific to young adults.

For programs delivered via digital channels with limited synchronous interactions, participant safety monitoring is critical. For those tailored participants, study staff were alerted to weight losses (or gains) that exceeded three pounds within a one-week period. Furthermore, all participants completed monthly health screens by responding to text messages and all weight loss participants were asked about their rate of weight loss or gain via text. Study staff monitored the Facebook groups for inappropriate or offensive content which was removed or responded to.

At 6 months, there was no overall effect of study group for weight loss. There was a moderating effect such that among those with the lowest BMIs (25–27.5 kg/m^2^), participants who were assigned to the tailored group lost 2.7 kg [−3.86, −0.68] and those in the targeted group lost 1.72 kg [−3.16, −0.29] more than those in the control group after adjustment for covariates [34].

What qualifies as a critical threshold for engagement in digital interventions has yet to be established [30]. Furthermore, frameworks and methods for evaluating the impact of Facebook posts also need to be standardized (Arnold CE reference) [35]. One such framework for examining the impact of messages is McGuire’s Model of Communication and Persuasion [36], which can be used to examine characteristics of posts (e.g., informational, lesson-based, poll, study-generated or user-generated). The effect of each post can be calculated using a traditional Facebook engagement equation of impact per post = [(Total Engaged Users / Total Reach) × 100] [37]. Encouraging researchers to proactively plan to collect and measure these characteristics will help future researchers design and implement interventions with effective messaging.

#### 3.1.2. Use Case #2: Nigeria Social Media to Promote COVID-19 Vaccination Impact Evaluation

The Bill & Melinda Gates Foundation (BMGF) has the goal of immunizing at least 500 million health care providers (HCPs) and high-risk people with COVID-19 vaccines worldwide. By achieving this goal, the foundation seeks a return to routine immunization, maternal and child health, and reproductive health services, which is critical to the COVID-19 response and to the strengthening of health systems. This effort focuses especially on low- and middle-income countries (LMIC), which have been disproportionately impacted by the COVID-19 pandemic. Nigeria, the largest country by population in Africa, is an important focus of these efforts.

Recently, BMGF sponsored a social media campaign in Nigeria to promote COVID-19 vaccination, with a primary focus on health care providers (HCP). The theory of change (ToC) of this campaign posits that HCP serve as crucial role models to encourage patients and the broader population to vaccinate. The campaign engaged a widespread group of health and other public-facing organizations, including the private and public sectors, to help design and deliver posts on Facebook, Instagram, and other platforms to promote vaccination. A series of campaigns using these social media platforms ran in 2022 in six targeted states within Nigeria.

In a companion project, Evans and colleagues developed an impact evaluation to evaluate the performance of an associated BMGF investment; Reducing Vaccine Hesitancy among HCP in Nigeria, which is a series of social media behavior change campaigns to reduce vaccine hesitancy amongst HCPs. The impact evaluation aims to determine which campaigns and strategies are effective in reducing vaccine hesitancy among HCPs and at what level of cost-effectiveness. The evaluation analyzes the campaign ToC, based on the 5 “C’s” model of vaccination promotion [38], co-developed by Dr. Evans and the campaign implementation team in order to understand the processes and effects of social media in influencing vaccine hesitancy among HCPs.

The evaluation examines the mediating effects of social media engagement metrics and changes in norms and related beliefs about the efficacy and safety of vaccines on COVID-19 vaccine hesitancy, intentions, and uptake. The analysis uses structural equation modeling (SEM) and multi-level modeling (MLM) techniques to test the campaign’s ToC for mediation based on changes in attitude, beliefs, and norms about vaccination, and for moderation of environmental, personal characteristics of HCPs, and other social ecological factors on the intervention’s effectiveness.

This study uses a novel social media platform for recruitment and data collection, Virtual Lab LLC (https://vlab.digital/ accessed on 22 May 2022), a data collection platform and a tool to enable study design. This platform enables researchers to create bespoke panels of respondents (i.e., a group of participants meeting evaluation eligibility criteria who agree to complete a baseline (BL) survey and to be re-contacted for follow up surveys). Participants may be recruited specifically from the target audiences for a behavioral intervention, such as a health campaign, and stratified by individual-level covariates of interest (e.g., geographic locations where the intervention is conducted, being a certain age, gender, race/ethnicity, or other demographic) and other relevant variables identified by researchers.

The use of the Virtual Lab platform for this study is a major innovation and will advance the field of digital media intervention research. The project will demonstrate (1) how social media recruitment and data collection may be effectively used to support social media interventions, especially as they focus on vaccination promotion, and (2) will demonstrate multiple evaluation strategies including geographic cluster comparisons as well as individual-level randomized controlled trials.

The materials developed in the project include surveys, interview guides, and social media content and metrics on dosage/exposure to the vaccination campaign at the population level. Researchers are compiling a library of such materials to be shared publicly, which will contribute to the design and development of future programs and campaigns, as well as increase the evidence for what works in digital health interventions worldwide.

The project includes substantial data collection, with some 8500 surveys completed in a baseline time point, prior to the campaign in late 2021. Additionally, 4 rounds of in-depth qualitative interviews will be conducted with some 30 social media outreach organizations and influencers that deliver the campaign in Nigeria. These data will be a major resource for building the evidence base on the effectiveness of social media for vaccination promotion and reducing vaccine hesitancy. This dissemination of findings and methodologies will make a substantial contribution to the social and behavioral sciences communities and the field of digital health [39].

#### 3.1.3. Use Case #3: Randomized Controlled Trial to Assess the Efficacy of Facebook Groups for COVID-19 Vaccination Uptake

Another example is a study aimed at promoting COVID-19 vaccination on Facebook. At the time of the study, over 35% of US adults had not received the COVID-19 vaccine [40]. While much had been published about vaccine misinformation on social media, little was known about how to use social media as an intervention platform.

Participants were eligible if they were adults who lived in the US and had not been vaccinated for COVID-19 (not even one dose) and had a Facebook account. While initially we planned to recruit on Facebook, recruitment on Facebook proved to be expensive and we soon shifted to recruiting on Amazon Mechanical Turk (Amazon MTurk). We recruited unvaccinated individuals (N = 353) and randomized them into one of two private Facebook groups (intervention, control) and measured their attitudes and behaviors at baseline and 2, 4, and 6 weeks after enrollment.

The intervention group consisted of novel educational content promoting the COVID-19 vaccine that was delivered daily for 4 weeks and sponsored by the GW Health Communication Corps, while the control group consisted of a referral to the COVID-19 Information Center on Facebook. In the intervention group, group moderators posted twice daily with information about the threat of COVID-19 in the US, and on the efficacy and safety of the COVID-19 vaccine (See Figure 3). Group moderators also used polls and posts to engage group members and encouraged comments and group discussion. Furthermore, moderators took on the role of replying to comments and questions raised by group members and were trained to use an empathic and non-judgmental tone. We compared the efficacy of these strategies across groups on vaccine acceptance, hesitancy, and uptake. In additional waves of the study, we also examined the role of group size and the use of group moderation features (e.g., allowing posting by group members) on group functioning (e.g., violating the group rules) and outcomes of interest. This study demonstrated that a new intervention format and platform for promoting vaccine uptake was feasible. Participants are being followed-up longitudinally to assess the efficacy of the intervention.

#### 3.1.4. Use Case #4: Pilot Randomized Controlled Trial to Evaluation the Efficacy of Anti-Vaping Advertisements

There is a dearth of published research on the effectiveness of social media interventions for behavior change in tobacco control [19]. The objective of a study led by Evans and colleagues was to determine the feasibility of the Virtual Lab platform to recruit participants and assess awareness of an anti-e-cigarette health campaign on Facebook. For this pilot feasibility study, researchers aimed to recruit 300 participants through Facebook using the Virtual Lab platform [14]. In order to recruit participants, we created recruitment ads and delivered them through the Facebook page we created called “Digital Media Experiment”. In order to demonstrate the credibility of the page, we posted related content and acquired likes. We used this Facebook page to run recruitment ads in August 2021. The recruitment ads were served to our target audience which included men and women 18–24 years of age who are located in the United States. After recruiting participants, we created another Facebook page called “Consumer Consciousness”. This page was used to run the target ads on the recruited participants’ Facebook newsfeeds during the study time period. We created this second Facebook page, with a different name, to control for any bias. Although we created two different Facebook Pages, they were both under the same Facebook Business Account, “Digital Health Research”.

The pre-test and post-test surveys were completed using a survey platform called Typeform. Surveys were linked to our Facebook pages using the Virtual Lab interface. This platform allowed our Facebook pages to send automated messages through Facebook Messenger to participants who clicked on the recruitment ads. The two recruitment ads used in this feasibility study were designed using 99design.com. The ads used the text “Take a 15 min survey, get paid $10”. See Figure 4. After participants clicked on the study’s ad, they were sent a message via Facebook Messenger inviting them to participate in the study. Before beginning the pre survey, participants were sent messages regarding the topic of the survey, compensation, privacy measures regarding collected data, and contact information if they had any questions regarding the study. Following those messages, the participants were sent a message asking if they would like to continue in order to consent to their participation in the study.

The study’s Facebook recruitment ads had a reach of 10,309 which is defined as the number of people who saw the ad at least once. The recruitment ad generated 15,718 impressions which is the number of times the ad was displayed on a person’s screen, and it was clicked on a total of 790 times. The Link-Click-Through Rate was 4.77% which is the percentage of times a person saw the recruitment ad and clicked on it. The study’s Facebook target ads had a reach of 191 people and generated 441 impressions. The target ads were played a total of 353 times and only 11 times where the ads played at 100% of its length.

The development of methods for experimentation within social media platforms is essential for the progress of public health media campaign research, especially in the context of tobacco control. This study demonstrated a new platform that allows for customized recruitment and longitudinal follow-up of participants, as well as the execution of survey research within Facebook, which was found to be feasible for media campaign awareness studies [14].

## 4. Discussion

The field of digital media for behavior change is growing rapidly, but research is still in its infancy. While a recent review found some 3300 digital media publications related to behavior change, only a fraction of these provided evidence for the effectiveness of a digital intervention in changing behavior. Many studies monitor the digital media landscape, such as “digital listening” projects, and studies of large social media datasets (infodemiology), which are critically important and advance our understanding of the digital landscape. However, rigorous intervention studies on the effectiveness of digital media in actually changing behavior, specifically in public health, remain relatively sparse [41]. The field has tremendous room for growth in the coming years.

Behavior change can take time, and the potential for regression to earlier states is well-known [42]. Future research should include longitudinal follow-up to assess the long-term effect of social media behavioral interventions. Additionally, there was a lack of evidence on the effectiveness of theories of change in social media interventions, and future research should focus on testing the processes of change.

Given the relative dearth of rigorously evaluated digital media interventions for behavior change, more formative research evaluating the feasibility, appropriateness, and acceptability of specific types of projects is needed. A more rigorous application of the principles of program evaluation will help develop targeted, effective digital media interventions.

In particular, digital media interventions need to explore the full range of functionality of digital devices and their near-constant role in personal self-management and day-to-day living to maximize opportunities for behavior change. One area that deserves further attention is the potential for multi-factorial studies that examine the effects of adding and subtracting features of digital devices (e.g., an intervention with and without an app, with and without social media interactivity, etc.) on behavior change. More elaborated research designs that examine how to optimize delivery of digital interventions using the full range of functions of devices such as mobile phones and tablets is needed.

One of the strengths of social media interventions is that objective dosage and exposure data from analytics are available. However, some studies have reported that their social media efforts were effective without clearly reporting quantitative data (e.g., clicks, shares, views, etc.) on social media use [1]. Future research should examine the characteristics of engagement exposure to evaluate dose-response effects—i.e., to determine whether more exposure or exposure of a specific type is associated with successful behavior change. This is important in order to be able to objectively attribute intervention effects to observed behavior changes and build the evidence base in the field.

Another important dimension of digital media for behavior change is health literacy. Digital media represent another dimension of health literacy: the ability to successfully use, navigate, and obtain benefit from health-related information on digital devices [43]. Research should focus on how to make digital media behavior change interventions more sensitive to health literacy needs, and on how to improve the extent to which participants can successfully consume and use digital health information.

Finally, future experimental research should rigorously examine the effects of variable levels of engagement with, and frequency and intensity of exposure to, multiple forms of digital media for behavior change. In particular, longitudinal studies that follow participants over extended periods of time are needed to evaluate more distal outcomes (e.g., beyond immediate content recognition, engagement, and short-term attitudinal outcomes. Studies should also investigate dose-response effects by increasing the number of digital ad exposures over an extended period of time and evaluating varying dose-response curves for different campaign outcomes [42]. Adding greater levels of digital exposure would allow a study to plot possible threshold and drop-off effects of exposure. Such research will inform our understanding of the impact of varying levels of digital ad exposure on longer-term outcomes.

## 5. Conclusions

This review provides an overview of the emerging field of digital media for behavior change. The field represents an important, relatively new domain that overlaps digital health, and the broader use of digital media for social change programs. While the evidence base in this field, as illustrated in this paper, is relatively small, the field is growing. Future research and programs should expand the domains of subject matter addressed by digital media for behavior change. More rigorous, controlled, and externally valid studies are needed and the field will need to stay current with rapidly emerging new digital technologies.

## Figures and Tables

**Figure 1 ijerph-19-09129-f001:**
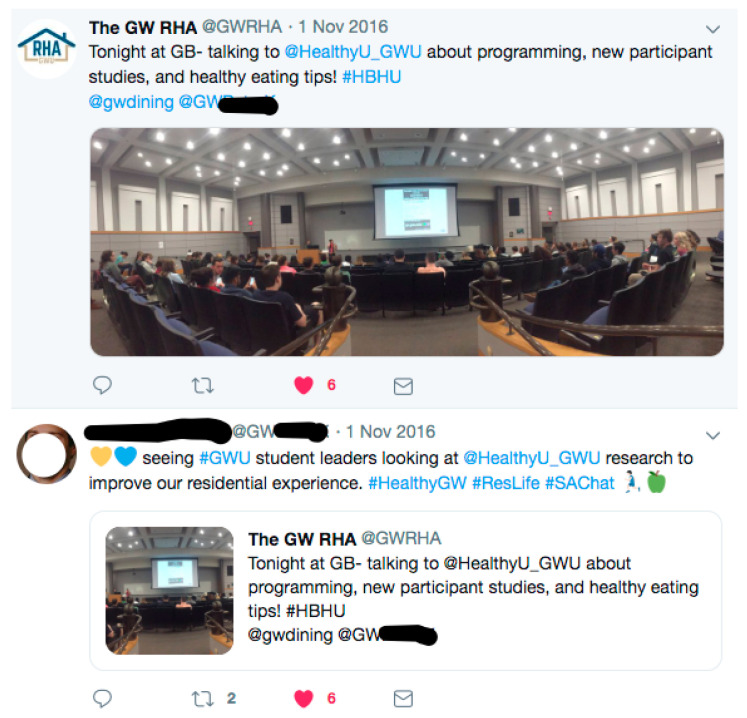
HBCU Facebook post 1.

**Figure 2 ijerph-19-09129-f002:**
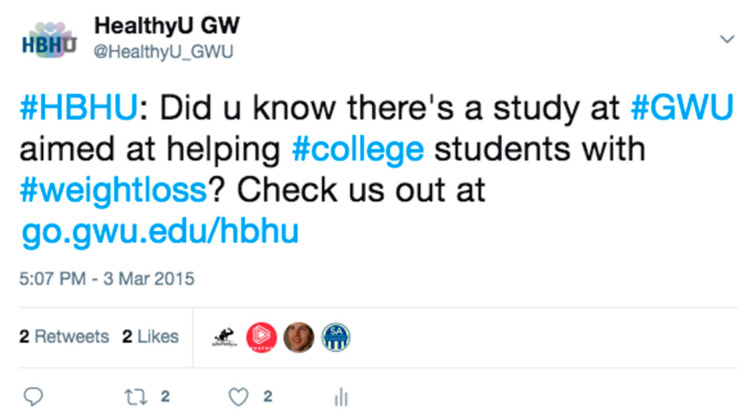
HBHU Facebook post 2.

**Figure 3 ijerph-19-09129-f003:**
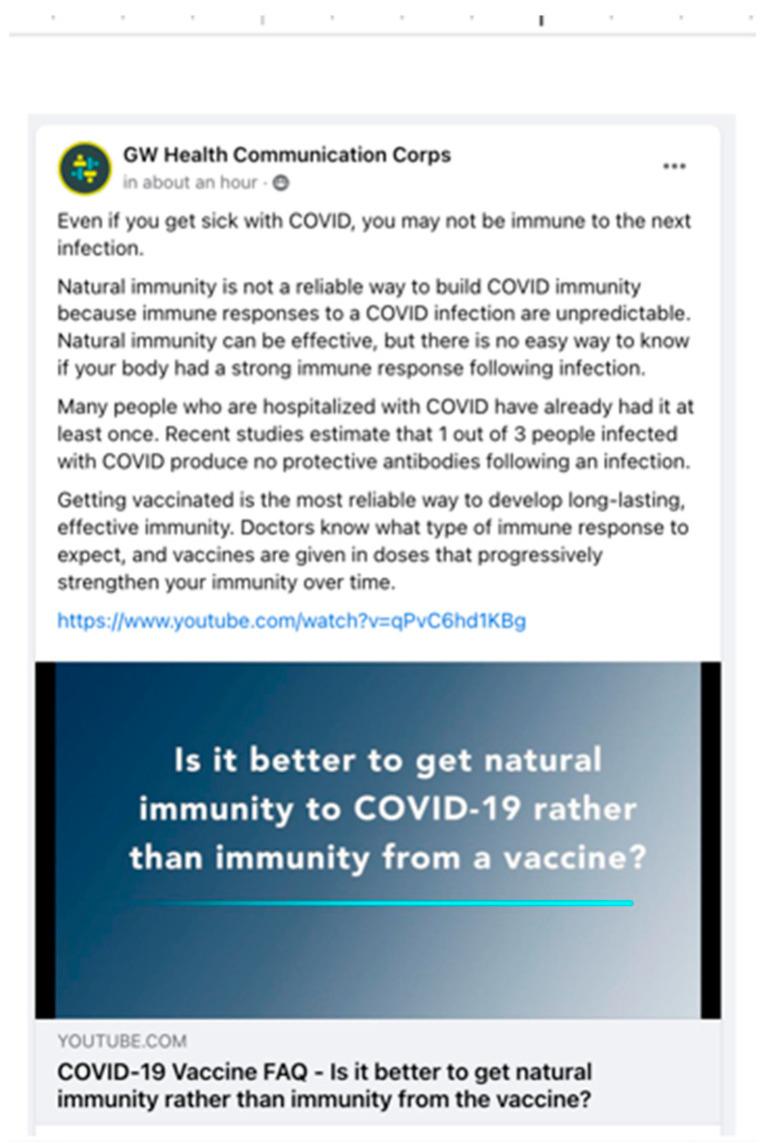
Sample post.

**Figure 4 ijerph-19-09129-f004:**
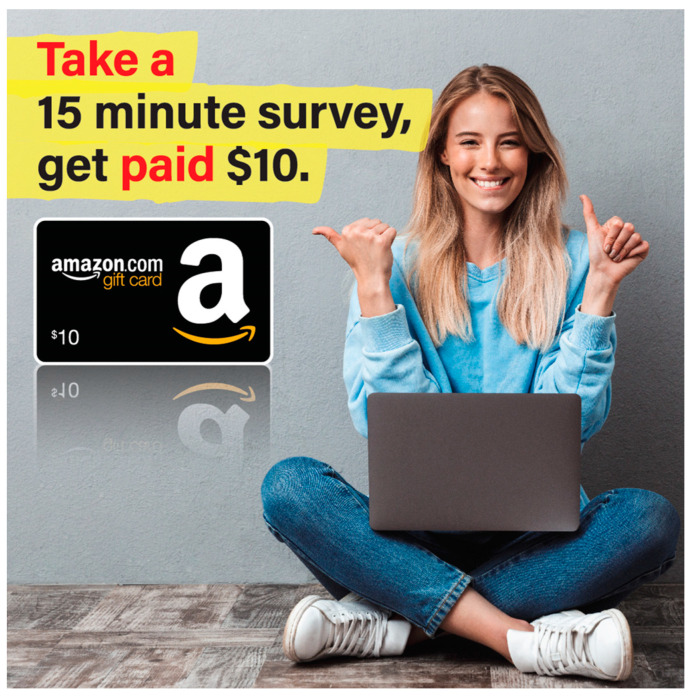
Facebook study recruitment ad.

## Data Availability

Not applicable.

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
