# Peer review of "Digital Media for Behavior Change: Review of an Emerging Field of Study"

_ijerph, 2022, doi:10.3390/ijerph19159129_

Round 1

Reviewer 1 Report

n/a

Author Response

We thank the reviewer for the review. We have made revisions to the paper to improve clarity and readability in response to other reviewer comments. As there are no specific comments and suggestions noted here, we thank the reviewer and will proceed with resubmission.

Reviewer 2 Report

Dear Authors,

1. First, the title of your paper should present that it is a literature review. This shall increase the visibility of your article.

2. What do you refer by ”describe a future agenda”? 

3. The template of the article, in terms of contents of every section, is not respected. You should provide more information in the Methodology section. I did not understand how you did this review. Please study in detail research presenting reviews (including PRISMA) and follow their structure as an example.

4. Examples shall be restructured in a separate section. 

5. The same with the case studies. Actually, which is the difference between examples and case studies and what is their scope in your work?

I look forward to checking your improved version.

Yours faithfully,

Author Response

We have attached a file with detailed responses. Thank you.

Round 2

Reviewer 2 Report

Dear Authors,

I look forward to seeing your paper published.

Yours faithfully, 

This manuscript is a resubmission of an earlier submission. The following is a list of the peer review reports and author responses from that submission.

Round 1

Reviewer 1 Report

This is a well presented exploratory piece which does not over-claim its significance, being limited to a review of cases and an overview of an emerging field. 

Given this stated scope, there is little to criticise and the article clearly presents a range of intervention methods and example cases in category, before moving to a discussion of the 'known unknowns' - ie where further research is required to understand the nature of engagement with digital media in these contexts and the relationship between such engagements and behaviour change.

However, as that need is identified, that does expose a gap in this work, which we can attribute to the starting point - ie the field context, as this is so interdisciplinary and intersecting. There is a large body of work on digital media literacies and forms of engagement, civic, educational but also, emerging, health communication related, and some of this work includes theories of change for more precisely identifying the complex relationship between digital media, publics, engagement and behaviour. These studies sometimes use the capability model (from Sen) to investigate how digital and media literacies convert to capabilities and actions in the social work, including those related to health related behaviours. 

So, the article does what it sets out to do well, and can be accepted as such, but I would advise at least the acknowledgement of this kind of work as a sub-field which should be of value for the kinds of future enquiries the authors propose. 

Reviewer 2 Report

This paper is a narrative review of the methods in which digital media is employed for behaviour change within the health field. It is an interesting contemporary topic and I appreciate the authors work to try and thematically bring together the findings from a large body of literature in a broad and disparate field. While the authors have made a good attempt to provide a comprehensive overview, I do not think the paper in its current formation particularly adds anything new to the existing body of research in this area. The scope of the paper is too broad whilst the narrative and examples are too narrow to provide sufficiently meaningful insight into these issues.

I was a little uncertain about the purpose of the paper and feel it requires some clearer parameters.  I felt the terms including digital media, digital media interventions and the use of digital media in research needed to be more clearly defined. The initial introduction suggested this was a narrative review which covered the use of digital media in interventions and in formative and evaluative research. Firstly, this was quite a broad undertaking and I wonder if the paper would be tighter if it focused on just one of these aspects. In particular, I don’t think the section on digital media for research adds anything to the evidence base in its current state – I think it is too focused on recruitment to online surveys and requires a much more detailed and focused view on the issues specific to digital media to provide a useful perspective on this issue.

Secondly, the subheadings in the main body of the paper read more like a “how to guide” for researchers wanting to use digital media but the text doesn’t really reflect these headings – providing broad overviews rather than the concrete details of how to achieve these aims. For example, the section on how to deliver digital media campaigns presents a very short and quite sparse overview of what is in itself an extremely broad field which draws on an even longer history of mass media marketing.

I was also unclear about the “use cases” presented – it was not clear why these two examples were chosen, if and how they represent examples of best practice in the field, or how they relate to the broader paper aims (both are interventions, when much of the previous main body focused on research). There is also no accompanying narrative on digital media interventions more broadly, and I do not feel these case studies in isolation present a sufficient overview of this issue to meet the paper’s objectives. I also think the presentation of the case studies should be consistent, currently one uses a series of sub-headings whilst the other does not.

I also have a few more minor suggestions for the authors in relation to their paper

  • I would suggest that perhaps the title should be made more specific to include health/public health and that this is also included in the key words as this will make the paper more easily searchable
  • I also think it would be useful to reflect that the focus of the paper is on changing health behaviours in the introduction. I know this might seem a rather obvious point to those of us in health-related fields, but digital media is used to achieve other types of behaviour change (particularly in relation to marketing) and so the paper may draw a wider audience who do not take the health-related nature of the term behaviour change for granted.
  • I think a very brief definition of the term digital media would be beneficial in the first paragraph as it isn’t completely clear from this how digital media differs from social media and a tighter description in the second paragraph of the parameters for this narrative review. In paragraph 2 you discuss “strategies to change behaviour implemented using digital technology” but in the second paragraph is appears that your paper is more broad than that as it also includes digital media as a methodology for data collection and research.
  • “and as an aspect of the environment (i.e., the study of infodemiology) that may influence behavior and moderate the effects of interventions aimed at behavior change” – I found the meaning of this part of the sentence difficult to follow
  • The final paragraph of the introduction discusses the health impacts of social media as a justification for research but this is not the focus of the paper and so confused the narrative a little. Also not clear on the distinction between digital and social media.
  • Page 3 – the example on canvassing for votes seemed out of place here as not health related. Would be better to provide an example from the weight management papers cited.
  • The authors describe a previous scoping review but it was not clear if this paper was an extension of that review which presented the same data to achieve additional objectives or whether the current paper had used a different corpus of literature. If the literature for this paper was different, I was not entirely clear how the two pieces of work were distinct and what the current paper added that had not been achieved from the more comprehensive and systematic approach taken in the scoping review.
  • In the section on how to use digital media for formative research, the main example cited is a study looking at drivers of Covid-19 vaccine uptake among Nigerian health workers but I could not see any mention of using digital media in this example beyond an online survey (which I assume recruited via social media although this isn’t clearly stated). An online survey is a fairly standard method of data collection which does not seem to particularly show case the use of digital media in formative research.
  • References use different fonts.

Reviewer 3 Report

The topic that the Authors choose is important and up-to-date.

However, the article's title does not underline which sphere” the growth of a promising practice” is the concern.

Abstract part

the Authors wrote:

„ The purpose of the current paper is to describe methods in the field, present use cases, describe a future agenda, and raise central questions to be addressed in future digital health research for behavior change.”

Which, in my opinion, does not reflect the title's understanding.

Then

„We examine use cases across several content ….”

Introduction part

The justification for digital media used in society is correct. The authors discussed the importance of social media and apps as the difference that appears in high- and low-income countries. Further intervention strategies are mostly related to social media and their media effects.

Authors analysed 3,300 articles whin in 298 possed the digital behaviour change intervention

Methods

In the methods, part Autors do not write about their methods in the article review.

The mentioned three main methods show the division of topic classification – or used digital media tools.

Like:

digital media interventions aimed at behavior change,

- social media (e.g. Facebook, Twitter) – three articles are mentioned [20,21,22]

formative research using digital media aimed at aiding in the design of interventions,

- Nigeria study in 2021 – one reference to Ahga

digital media used to conduct outcome and impact evaluations

- Virtual Lab, Facebook Messenger, - one reference [23]

This part should be more underlined in reached literature overview.

Use cases (page 5)

Figure 1 – what is the scientific sense of publishing a Facebook post?

Intervention descriptions - Page 7 – initially, the 298 interventions were mentioned, but the Autohgs finally only presented two of them  [32,22]. I am not aware of how they chose them as they do not include any methods.

The most rebuilt part includes the case of Nigeria description (case 1).

The second case 2 is the promotion of covid vaccination – one reference were mentioned [40]

In my opinion, the discussion part is limited and includes only two references? How to discuss achieved results?

Based on 298 articles and only 41 references, the article does not properly express the underlined topic. It will be good to add the table with systematic literate review results.

I do not recognise the directed method limitation of the study.

The Authors, however, include further research directions.

Reviewer 4 Report

Dear Authors,  Thank you for submitting your paper to IJERPH. I read it with great interest. I believe that the topic you have chosen is relevant, and it relates to essential issues in our lives. Unfortunately, it is far too early at this stage for your paper to be published. Apart from numerous linguistic and editing errors, which you should have eliminated at the initial submission stage, the study suffers from significant methodological shortcomings and deficiencies in terms of content. Allow me to list them for the sake of clarity of my position. - The title is misleading because the part referring to 'Growth of a Promising Change' is not entirely clear from your text. This may be due to the inadequate design of the empirical part of your study. In any case, it is difficult to grasp this "growth", its causes, symptoms and effects - the 'Introduction' part should be summed up as 'Introduction and literature review, as it contains both introductory information and numerous references to the literature - the phenomena of "digital interventions" and "digital health" should be defined since references to "normal" health (weight, vaccines) appear later in the text. Instead, the term "digital" suggests focusing on digital well-being, digital hygiene, etc. So using it without a precise explanation can be misleading - it is not clear to me why for the "Digital Health Research", you paid so much attention to marketing and why you treated the DH literature selectively, focusing only on how to study it - "Evidence on Digital Media..." lacks a definition of digital media. Besides, there is no evidence of the impact of digital media on behaviour change (neither does the health issue to which you referred earlier) - in two different places of the test, you indicate your research objectives, but there are no research questions and hypothesis - you do not describe your research method precisely. Was it a literature review (you suggest so on p. 3)? If so, the information given is too vague and does not describe the research methodology. Where did you look for texts, and in which scientific databases? How did you do it? What were the subsequent steps in selecting a suitable research sample? Why do you not use the PRISMA method if you write about "systematic review"? How did you analyse the final sample, and with what tools? Who studied the selected texts, and when? Did you use cross-checking? - Do you list three identified methods for researching your chosen topic? Are these your terms? On what basis do you build them, and on what data do you introduce such systematics? The idea is that you should obtain quantitative and qualitative data from the systematic literature on the subject. These should then be presented in a form, preferably % or numerative, so that the reader can see that the conclusions drawn are based on a multi-stage data analysis. In other words, the three methods result from a study of almost 300 texts you have selected. Meanwhile, we have several studies attached to each method and thus do not know where these methods came from. - On what basis did you choose the case studies discussed? What made you choose them? Was it due to, e.g. the number of citations and the popularity of the research topic? In addition, references to them are descriptive and reproductive - there is no clear explanation of how they relate to the theme of 'Growth from a Promising Change'.  - The discussion is based on generalisations. Again the conclusions do not follow from the text. Recommendations for future research, which should result from the identified research gaps, are currently in the form of assumptions rather than substantively justified postulates. There is also no indication of the limits of your research.   Considering the above numerous remarks (including mainly the lack of a properly developed methodology), I conclude that the paper cannot be published in its present form. For this to happen, its design requires considerable work and, in some places, a complete redesign. I am very sorry to bring such bad news. Sincerely.